# S100B Maternal Blood Levels in Gestational Diabetes Mellitus Are Birthweight, Gender and Delivery Mode Dependent

**DOI:** 10.3390/ijerph19031028

**Published:** 2022-01-18

**Authors:** Laura Abella, Ebe D’Adamo, Mariachiara Strozzi, Joan Sanchez-de-Toledo, Miriam Perez-Cruz, Olga Gómez, Ernesto Abella, Maurizio Cassinari, Roberto Guaschino, Laura Mazzucco, Antonio Maconi, Stefania Testa, Cristian Zanelli, Marika Perrotta, Patacchiola Roberta, Neri Costanza Renata, Giorgia Gasparroni, Ester Vitacolonna, Francesco Chiarelli, Diego Gazzolo

**Affiliations:** 1Hospital Universitari Dexeus, 08028 Barcelona, Spain; laura.abelladlt@gmail.com; 2Neonatal Intensive Care Unit, G. d’Annunzio University, 65100 Chieti, Italy; ebe.dadamo@asl2abruzzo.it (E.D.); perrottamarika@hotmail.it (M.P.); giorgia.gasparroni@gmail.com (G.G.); 3Neonatal Intensive Care Unit, ASO SS Antonio, Biagio, C. Arrigo, 16100 Alessandria, Italy; mstrozzi@ospedale.al.it (M.S.); mcassinari@ospedale.al.it (M.C.); rguaschino@ospedale.al.it (R.G.); lmazzucco@ospedale.al.it (L.M.); amaconi@ospedale.al.it (A.M.); stesta@ospedale.al.it (S.T.); czanelli@ospedale.al.it (C.Z.); 4Fetal Medicine Research Center, Hospital Sant Joan de Deus, University of Barcelona, 08950 Barcelona, Spain; SanchezDeToledoJ@upmc.edu (J.S.-d.-T.); lgomezroig@sjdhospitalbarcelona.org (M.P.-C.); OGOMEZ@clinic.cat (O.G.); 5Hospital Dr. Max Peralta Jimènez, Cartago 30101, Costa Rica; info@drernestoabella.com; 6Department of Pediatrics, G. d’Annunzio University, 65100 Chieti, Italy; robertapatacchiola92@gmail.com (P.R.); costanzanerix@gmail.com (N.C.R.); chiarelli@unich.it (F.C.); 7Department of Medicine and Aging, School of Medicine and Health Sciences, G. d’Annunzio University, 65100 Chieti, Italy; ester.vitacolonna@unich.it

**Keywords:** biomarker, brain development, fetus, newborn, pregnancy, S100B, gestational diabetes mellitus

## Abstract

Gestational Diabetes Mellitus (GDM) is one of the main causes of perinatal mortality/morbidity. Today, a parameter offering useful information on fetal central nervous system (CNS) development/damage is eagerly awaited. We investigated the role of brain-protein S100B in the maternal blood of GDM pregnancies by means of a prospective case–control study in 646 pregnancies (GDM: *n* = 106; controls: *n* = 530). Maternal blood samples for S100B measurement were collected at four monitoring time-points from 24 weeks of gestation to term. Data was corrected for gender and delivery mode and correlated with gestational age and weight at birth. Results showed higher (*p* < 0.05) S100B from 24 to 32 weeks and at term in GDM fetuses than controls. Higher (*p* < 0.05) S100B was observed in GDM male new-borns than in females from 24 to 32 weeks and at term, in GDM cases delivering vaginally than by caesarean section. Finally, S100B positively correlated with gestational age and weight at birth (R = 0.27; R = 0.37, respectively; *p* < 0.01). The present findings show the usefulness of S100B in CNS to monitor high-risk pregnancies during perinatal standard-of-care procedures. The results suggest that further investigations into its potential role as an early marker of CNS growth/damage in GDM population are needed.

## 1. Introduction

Gestational diabetes mellitus (GDM) is the perinatal complication showing the greatest increase, occurring in up to 15% of healthy pregnancies [1]. GDM still represents one of the main causes of perinatal mortality and morbidity [1,2]. Perinatal multiorgan complications regard fetal-neonatal growth (i.e., macrosomia, growth restriction), respiratory distress syndrome, metabolic and ion disorders, and brain development/damage [2,3,4].

Nowadays, the monitoring of insults and contributing factors in high-risk pregnancies is restricted to: (i) adequate documentation of general-medical and obstetrical factors, (ii) laboratory and screening tests, (iii) fetal-neonatal Doppler and ultrasound monitoring, and (iv) post-natal laboratory analysis, blood pH and neonatal neuroimaging [5,6].

Recently, it has been suggested that the assessment of neurobiomarkers (NBM) in biological fluids could offer useful information on central nervous system (CNS) development and damage when standard monitoring tools may still be silent or unavailable [7,8]. Among NBM currently investigated in the perinatal period, S100B, a CNS-specific acidic calcium binding protein mainly expressed in glial cells and neurons seems to be the most promising, especially due to: (i) brain-specific site of concentration (glial cells, neurons), (ii) trophic and brain damage marker functions [7,8,9,10,11,12], and (iii) measurability in different biological fluids (cerebrospinal, blood, amniotic, urine, saliva, milk), particularly the maternal blood [7,13,14,15,16,17,18,19,20]. This is significant since maternal blood constitutes the best option for monitoring fetal-maternal well-being in the absence of any sampling stress for the fetus. Increased S100B maternal blood levels have been found in growth-restricted fetuses developing postnatal intraventricular hemorrhage [21]. In healthy and pregnant women carrying a fetus small for gestational age, a gradient between fetal and maternal bloodstreams has been found [22,23]. More recently, a reference curve of the protein in the II-III trimester of pregnancy has been provided and S100B has been found to be gestational age (GA), gender and delivery mode-dependent [23]. However, data on S100B changes in women complicated by GDM is still lacking.

Therefore, in the present study we aimed at investigating whether S100B levels in maternal blood of GDM women differed from healthy pregnancies and correlated with fetal/neonatal growth parameters.

## 2. Materials and Methods

### 2.1. Population

Informed consent was obtained from all women before inclusion in the study, and approval was obtained from our Local Human Investigation Committees (CoMBINeASO.Neonat17.01).

Based on our epidemiological data reporting about 5000 deliveries per year admitted to our III level referral center for obstetrics and neonatal care, we conducted a prospective study from July 2016 to December 2020.

For the calculation of sample size, we used S100B changes as the main parameter. As no basic data are available for the studied population, we assumed a change in 0.5 SD in S100B in maternal blood to be clinically significant. Indeed, considering α = 0.05 and using a two-sided test, we estimated a power of 0.90 recruiting 96 high risk patients per group. We added *n* = 10 cases per group to allow for any dropout. Therefore, we set out to recruit a total of 106 pregnant GDM-complicated women. The control group included 530 healthy pregnant women (1 GDM vs. 5 controls) (Figure 1).

Healthy pregnancies and new-borns were classified according to the criteria of The American College of Obstetricians and Gynecologists (ACOG) and the American Academy of Pediatrics (AAP). In detail, GA was determined by the last menstrual period and confirmed by a first trimester ultrasound scan. Appropriate growth was defined by the presence of ultrasonographic signs (biparietal diameter and abdominal circumference between the 10th and 90th centiles), according to the nomograms of Campbell and Thoms, and by post-natal confirmation of a birthweight (BW) between the 10th and 90th centiles, according to our population standards, correcting for the mother’s height, weight, parity, and the sex of the new-born [24,25]. At birth, new-borns fulfilling all the following criteria were classified as normal: no maternal illness; no signs of fetal distress; pH > 7.2 in cord or venous blood; and Apgar scores > 7 at 1 and 5 min.

GDM was classified according to national and American Diabetes Association guidelines. A 2 h oral glucose tolerance test (OGTT) with a 75 g oral glucose load was performed. The diagnosis was made when plasma glucose concentrations met or exceeded the following glucose levels: fasting levels of 92 mg/dL, 1 h of 180 mg/dL, 2 h of 153 mg/dL. The test was performed between 24–28 GA (mean 26 ± 2 weeks) [26]. Sixty-seven out of 106 GDM pregnancies were treated with insulin therapy and 34 out of 106 required diet therapy.

In all women, according to national guidelines, after GDM diagnosis maternal blood was drawn from the cubital vein for standard laboratory investigations and S100B protein assessment at four pre-determined monitoring time-points (T1: 24–28 GA; T2: 29–32; T3: 33–37 GA; T4: >37 GA). We included only pregnant women in whom at least three collection times occurred (T1: *n* = 106; T2; *n* = 102; T3; *n* = 100; T4: *n* = 99).

Exclusion criteria were multiple pregnancies, intrauterine growth retardation, gestational hypertension, obesity, fetal malformations, chromosomal abnormalities, perinatal asphyxia and dystocia.

### 2.2. S100B Measurement

Serum maternal blood samples collected at different time-points were immediately centrifuged at 900× *g* for 10′ and supernatant stored at −80 °C until assessment. The S100B protein concentration was measured in all samples using a commercially available immunoluminometric assay (Liaison S100, Dietzenbach, Germany). The limit of detection of the assay was 0.02 μg/L. The calculated within-assay and inter-assay variation was <10% and <5%, respectively.

### 2.3. Neurological Examination

Neurological examination was performed daily during hospital stay. Neonatal neurological conditions were classified using a qualitative approach as described by Prechtl, assigning each infant to one of three diagnostic groups: normal, suspect, or abnormal [27]. An infant was considered to be abnormal when one or more of the following neurological syndromes were present: hyper-or hypokinesia, hyper- or hypotonia, hemi-syndrome, apathy syndrome, and hyperexcitability syndrome. An infant was classified as suspect if only isolated symptoms were present but no defined syndrome.

### 2.4. Monitoring Parameters

On admission to the Neonatal Intensive Care Unit (NICU), all new-borns routinely underwent an assessment of laboratory parameters such as venous blood pH, carbon dioxide (PaCO_2_) and oxygen partial pressures (PaO_2_), base excess (BE), red blood cell count (RBC), hemoglobin concentration (Hb), hematocrit rate percentage (Ht), glycaemia, ion concentration.

## 3. Statistical Analysis

S100B concentrations in maternal blood were expressed as the median and 25th–75th centiles. Data were analyzed for statistically significant differences between groups by the Mann–Whitney U two-sided test and ANOVA followed by the Dunn post hoc when not normally distributed. Comparisons between proportions were performed by Chi-square test. Linear regression analysis for S100B and neonatal growth parameters correlation was performed. Statistical significance was set at *p* < 0.05.

## 4. Results

In Table 1, maternal and perinatal characteristics of the studied groups are reported.

No significant differences (*p* > 0.05) were observed in the studied groups regarding maternal age, delivery mode, Apgar scores at 1 and 5 min, blood pH and sodium levels. GA, BW, pCO_2_, pO_2_, BE, RBC, Hb and Ht and potassium blood levels although results output were within normal ranges for our study-population, significantly differed between groups (*p* < 0.001, for all).

As expected, glucose and calcium blood levels significantly differed (*p* < 0.05, for both) between groups as well as the CNS pattern characterized by isolated tremors and hyperexcitability symptoms requiring early feeding and infusion. All new-borns were in good clinical conditions and no overt neurological abnormality was observed at discharge from NICU.

### 4.1. S100B Protein Measurements

S100B protein levels were measurable in all samples collected. In healthy controls, the pattern of the protein in the II-III trimester of pregnancy fitted previous data [23]. When S100B levels were compared between the studied groups higher (*p* < 0.01, for all) protein levels were observed in GDM than controls at T1, T2 (24–32 GA), and T4 (>37 GA), whilst no differences (*p* > 0.05) at T3 (33–37 GA) were found (Figure 2).

#### 4.1.1. S100B and Gender

In healthy controls, no significant differences (*p* > 0.05, for all) in S100B levels were found between male and female sub-groups at II–III trimester time-points (data not shown).

In the GDM group higher (*p* < 0.01, for both) S100B levels were found in males than in females at T1 and T2, whilst no differences (*p* > 0.05, for both) were observed at T3 and T4.

When we compared the studied groups, after correction for gender, we found higher (*p* < 0.05, for all) S100B levels in GDM male offspring than in male controls at T1, T2 and T4. No differences (*p* > 0.05, for all) were observed between groups at T3 as well as between GDM female and female controls at T1–T4.

#### 4.1.2. S100B and Delivery Mode

In healthy controls, the pattern of the protein in the II-III trimester of pregnancy fitted previous data (data not shown) [23].

In the GDM group, higher (*p* < 0.01) S100B levels were found at T4 in cases delivered vaginally than in those by caesarean section whilst no differences (*p* > 0.05, for all) were found between sub-groups at T1–T3. When we compared the studied groups, after correction for delivery mode, we found higher (*p* < 0.05) S100B levels at T4 in GDM than in healthy vaginally delivered controls. No significant (*p* > 0.05, for all) differences were observed between sub-groups at T1-T3. Moreover, higher (*p* < 0.05) S100B levels at T4 were found in healthy than in GDM caesarean section delivered sub-groups. No significant (*p* > 0.05, for all) differences were observed between sub-groups at T1-T3.

Finally, significant positive correlations (*p* < 0.01, for both) between S100B levels at T4 and GA and BW were found (R = 0.27; R = 0.37, respectively) (Figure 3).

#### 4.1.3. S100B and Neurological Outcome

In the transition phase, according to Prechtl’s test results we sub-grouped GDM infants as follows: GDM suspect (GDM S), GDM normal (GDM N) and controls (C). Higher (*p* < 0.01, for all) S100B levels were observed at T1-T4 in GDM S when compared with GDM N and C, respectively. No differences (*p* > 0.05, for all) at T1-T4 have been found between GDM N and C (Figure 4).

## 5. Discussion

Among maternal pregnancy diseases, GDM incidence is rapidly increasing and hyper-glycemia is still one of the major causes of unfavourable perinatal outcome, affecting fetal/neonatal growth as well as CNS development [2].

Over the last decade, despite technological advances in the management of high-risk pregnancies, monitoring the well-being of the fetal-maternal dyad is still an issue which is far from being solved.

In the present study, we found that S100B maternal blood levels in GDM pregnancies, in the II-III trimester GA, increased more than in healthy controls and were also gender-and delivery mode-dependent. Furthermore, protein levels positively correlated with weight and gestational age at birth.

S100B pattern in the maternal blood of healthy pregnancies was in line with previous observations and offers additional support to fetal CNS monitoring by the measurement of brain biomarkers in the best non-invasive biological fluid [23]. Notably, the higher S100B levels in GDM pregnancies, from 24 GA to term, were in line with previous reports in high-risk pregnancies [21,22,28,29].

The finding of higher S100B maternal blood levels, in relation with perinatal growth and hyperglycemia in GDM women, warrants further consideration. In particular: (i) S100B can be released as a neurotrophic factor contributing to macrosomia [9,10,11,12]. The explanation resides in the increase in transplacental glucose transfer altering fetal metabolism, triggering fetal hyperglycemia and hyperinsulinemia responsible for macrosomia and altered tissue growth (i.e., adipose tissue, heart and skeletal muscle, liver, islets of Langerhans) [30], (ii) hyperglycemia can influence S100B release in the extracellular space, causing CNS cells damage/death through NO-synthase activation or apoptosis mechanisms [31], and (iii) saltatory/irregular umbilical cord compression episodes due to fetal macrosomia that can be responsible for hypoxic insults known to increase S100B in fetal and in maternal bloodstreams [32,33,34]. Overall, it may be argued that the higher fetal protein levels detected in GDM pregnancies are mainly related to altered glucose metabolism and macrosomia rather than perinatal hypoxia and CNS stress/damage. The finding is also corroborated by positive correlations between S100B and weight and gestational age at birth and is in line with previous observations in amniotic fluid when S100B correlated with ultrasound parameters suggestive of fetal and CNS growth (i.e., estimated weight, bi-parietal and transverse cerebellum diameters, head circumference) [9].

In the present series, we found that S100B is gender-dependent in healthy and GDM fetuses. The finding is in line with previous observations of higher female protein levels in healthy fetuses, new-borns, and children [9,10,35]. Conversely, in the GDM group, we found higher male S100B levels at 27–32 GA and no differences from 33 GA to term. The findings deserve further discussion in terms of: (i) different monitoring time-points between our data and previous observations (every week vs. four weeks, vs. before delivery) [21,22,23]; (ii) different fetal growth pattern in male GDM group in common with other clinical and anthropometric studies (i.e., height/weight growth reference curves), suggesting the possibility that brain maturation in males differed from females [10,11,35]; (iii) higher S100B levels in GDM than healthy males and no differences between GDM males and females from 33 GA to term suggesting faster male growth at the stages under investigation. Data in late preterm period is crucial for CNS maturation in terms of electrophysiology, brain weight and volume increase (35%; 37%, respectively) [36,37], synaptogenesis, dendritic arborization and axonal elongation processes, cerebral oxygenation and metabolism [38,39,40]. Overall, it is possible to argue that increased S100B levels in GDM males are supportive of an accelerated growth exerting increased fetal and putative CNS growth: however, the price to pay regards postnatal glucose metabolic alterations, CNS stress/damage and the risk of a higher rate of respiratory distress [2,3,4,5].

In the present study, we also found that S100B increased in infants showing minor neurological symptoms at birth. The finding is in agreement with previous observations in high-risk fetuses and infants [14,15,16,17,18,19,20,21,29]. Further studies in a wider population aimed at investigating the correlation between a well-established factor responsible of perinatal brain impairment such as hypoxia and metabolic alterations GDM dependent are so justified.

Lastly, we found that S100B in healthy GDM pregnancies is delivery-mode-dependent with higher protein’ levels in vaginally delivered cases. Results open the way to further investigations aimed at elucidating the role of the protein as a useful tool for determining the best timing for delivery.

Finally, we recognize that the present study has several limitations, such as: (i) the small GDM population recruited, (ii) the absence of S100B correlation with longitudinal glucose monitoring and insulin treatment [41], and (iii) long term follow-up. Further investigations aimed at addressing the aforementioned issues are eagerly awaited.

## 6. Conclusions

In conclusion, the present results suggest that S100B measurement in maternal blood can offer useful information on fetal CNS development/damage of high-risk cases. Data opens the way for further investigations aimed at clarifying the putative protein’s role as an early tool for the best timing of delivery.

## Figures and Tables

**Figure 1 ijerph-19-01028-f001:**
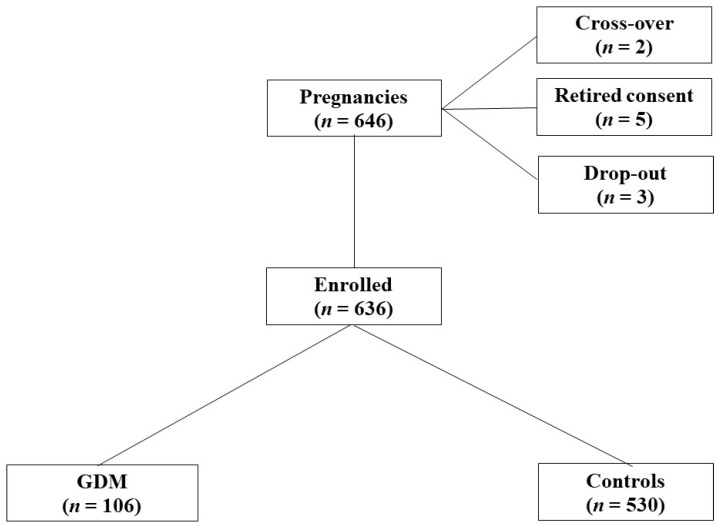
Flow chart showing patient recruitment.

**Figure 2 ijerph-19-01028-f002:**
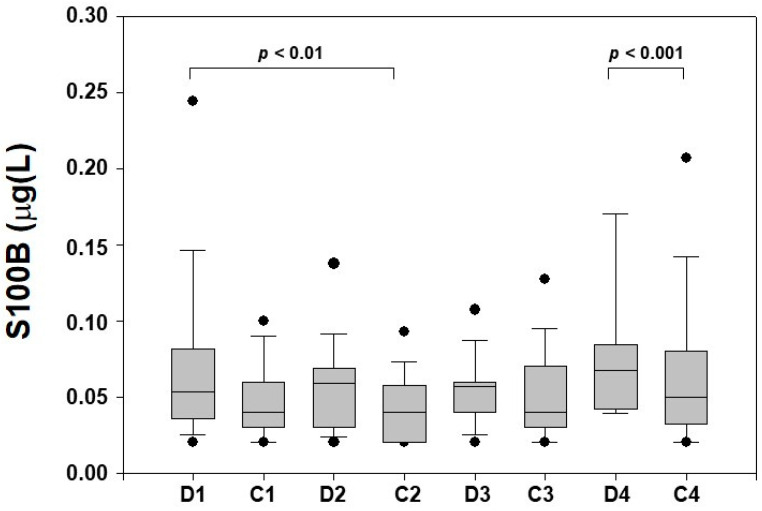
S100B maternal blood levels (μg/L) measured at different monitoring time-points (1: 24–28 weeks; 2: 29–32 weeks; 3: 33–37 weeks; 4: >37 weeks) in gestational diabetes mellitus pregnancies (D) and healthy controls (C). Data is given in median and interquartile ranges and 5th−95th centiles (●).

**Figure 3 ijerph-19-01028-f003:**
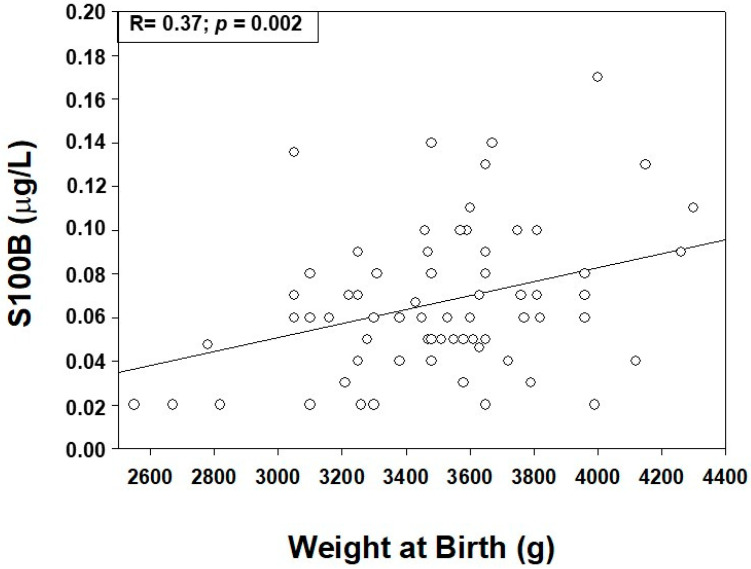
Positive linear regression correlation between S100B maternal blood levels (μg/L) measured at term, and weight at birth.

**Figure 4 ijerph-19-01028-f004:**
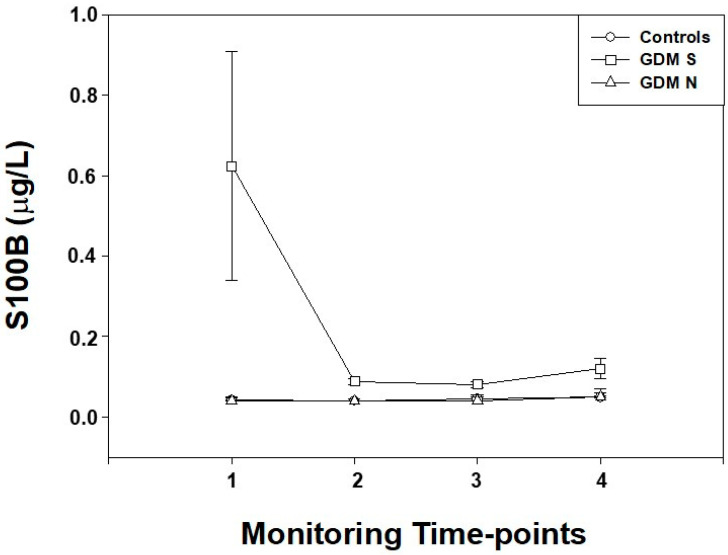
S100B maternal blood levels (μg/L) measured at different monitoring time-points (1: 24–28 weeks; 2: 29–32 weeks; 3: 33–37 weeks; 4: >37 weeks) in gestational diabetes mellitus pregnancies (GDM) whose new-borns showed a Prechtl score suspect (S, □), normal (N, ⌂) and in controls (C, ○). Data is given in median and interquartile ranges and 5th–95th centiles.

**Table 1 ijerph-19-01028-t001:** Perinatal outcomes and laboratory parameters recorded at birth in the gestational diabetes (GDM) and control groups. Data are given as means ± SD. * *p* < 0.05.

Parameters	GDM (*n* = 106)	Controls (*n* = 530)	*p*
Maternal age, (years)	24.7 ± 3.9	25.6 ± 4.4	NS
Delivery mode, *n*/total			
Caesarean section	38/106	117/530	NS
Vaginal	58/106	413/530	NS
Gestational age (weeks)	38 ± 2	39 ± 1 *	<0.001
Birth weight (g)	3019 ± 167	2902 ± 190 *	<0.001
Gender male/female	57/59	261/269	0.006
Apgar score > 7, *n*/total			
at 1 min	104/106	530/530	NS
at 5 min	106/106	530/530	NS
pH > 7.20, *n*/total	106/106	530/530	NS
pCO_2_, mmHg	41.2 ± 2.7	44.2 ± 2.1	<0.001
pO_2_, mmHg	43.2 ± 0.6	41.4 ± 1.8	<0.001
BE	3.2 ± 0.7	2.2 ± 0.9	<0.001
RBC count, 10^12^/L	4.1 ± 0.1	4.2 ± 0.2	<0.001
Hb, g/L	143 ± 3	142 ± 2	<0.001
Ht (%)	42.4 ± 0.5	41.5 ± 0.4	<0.001
Plasma glucose, mmol/L	2.2 ± 0.2	4.2 ± 0.3 *	<0.001
Na^+^, mmol/L	23 ± 0.7	23 ± 0.7	NS
Ca^++^, mmol/L	0.2 ± 0.03	0.4 ± 0.02 *	<0.001
K^+^, mmol/L	0.7 ± 0.06	0.8 ± 0.07	<0.001
Neurological examination			
Normal/ Suspect/Abnormal	70/36/0	530/0/0 *	<0.001

## Data Availability

Deidentified individual participant data will not be made available.

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
