# Peer review of "S100B Maternal Blood Levels in Gestational Diabetes Mellitus Are Birthweight, Gender and Delivery Mode Dependent"

_ijerph, 2022, doi:10.3390/ijerph19031028_

Round 1

Reviewer 1 Report

Thanks to the authors for giving much effort to make the manuscript attractive. But it would be nice if you would mark all the changes/modifications in the manuscript properly. Albeit it’s quite good, except few minor issues, e.g. on line 112, “2-h oral” and 75-g oral” should be “2 h oral” and “75 g oral” respectively. Please remove the hyphen sign before the units. There are several similar errors in the manuscript. Also, on line 129, “-80ºC” should be “-80 ºC”. Please give space before units as well.

Author Response

Please find attached about author's reply.

Reviewer 2 Report

I would like to thank the authors, they have satisfactorily addressed my comments raised on their original submission. From my point of view, it is now suitable for publication.

Author Response

Thank you so much for your comments.

This manuscript is a resubmission of an earlier submission. The following is a list of the peer review reports and author responses from that submission.

Round 1

Reviewer 1 Report

Manuscript: S100B Maternal Blood Levels in Gestational Diabetes Mellitus

are Growth, Gender and Delivery Mode Dependent

The authors have submitted a well-written and structured manuscript. The analysis and results presented are interesting. I believe that this study contributes to the scientific discussion on this topic. The data can serve as a reference for future research. I think it should be published. Minor comments follow.

  1. Materials and Methods section:
  • Please specify Inclusion and Exclusion criteria for your study. Also mention where these patients were enrolled.
  • Please provide information on the type of blood collection tubes used.
  • How long after sample collection did you complete specimen processing/analysis?
  1. Could you provide information on the management of gestational diabetes in these patients? Were they on insulin therapy?
  2. Page 6, Line 165: “GDM males” consider instead GDM male offspring.

Author Response

Reviewer 1

We want to thank reviewer for his kind comments and useful suggestions.

  • Please specify Inclusion and Exclusion criteria for your study. Also mention where these patients were enrolled.

Materials and methods section has been expanded according to reviewer’s suggestion. We trust that now is clearer.

  • Please provide information on the type of blood collection tubes used.

Materials and methods S100B measurement section has been expanded according to reviewer’s suggestion. However, Literature data provides evidence that the protein can be collected both in gel-coagulated or EDTA tubes in absence of any side-effects. Anyway we are available to include this information needed.

  • How long after sample collection did you complete specimen processing/analysis?

Materials and methods S100B measurement section has been expanded according to reviewer’s suggestion.We trust that now is clearer.

  • Could you provide information on the management of gestational diabetes in these patients? Were they on insulin therapy?

Materials and methods section has been expanded according to reviewer’s suggestion. We trust that now is clearer.

  • Page 6, Line 165: “GDM males” consider instead GDM male offspring.

We changed the paragraph according to reviewer’s suggestion.

Reviewer 2 Report

This quite interesting manuscript demonstrates the relationship between S100B and pregnancy outcome in GDM patients.  Although there are some issues, which need to address:

  • In the title you mentioned about “growth”, but in the study the attention on growth/weight gain is not strong enough. It would be better if you discuss more in depth about it. Otherwise, you can skip the word from title.
  • It would be better if you would kindly revise the manuscript by a native English speaker. Because there are several sentences with grammatical errors e.g., 2nd and last sentences of abstract, which are not clear.
  • What is the potential source of S100B protein?
  • Can S100B upregulation in maternal blood be due to GDM? Does S100B have any relationship with blood glucose level?
  • Did S100B level reduced following delivery?
  • What are the numbers of each gestational age groups among recruited population? Or all the recruited population of 24-28 GA? Please describe into method section (2.1) and figure 1.
  • Are the collected blood samples treated with anticoagulant? Did you measure S100B from serum or plasma? Please describe clearly on method section (2.2).
  • Have you measured amniotic fluid and/or neonatal S100B level? And is there any correlation with maternal blood S100B level? If possible, please include the data.  
  • Were the pregnant women with GDM taking insulin or any other medication? Any relationship with maternal blood S100B level? If possible, please include the data.

It’s nice to mention about the limitation of the study, which clarifies the depth of knowledge of the investigators. Thanks again for your wonderful effort to make a nice manuscript

Author Response

We want to thank reviewer for his kind comments and useful suggestions

  • In the title you mentioned about “growth”, but in the study the attention on growth/weight gain is not strong enough. It would be better if you discuss more in depth about it. Otherwise, you can skip the word from title.

According to reviewer’s suggestion we changed MS title. We trust that now is clearer.

  • It would be better if you would kindly revise the manuscript by a native English speaker. Because there are several sentences with grammatical errors e.g., 2nd and last sentences of abstract, which are not clear.

According to reviewer’s suggestion MS has been revised by our institutional native English speaker. We trust that now is clearer.

  • What is the potential source of S100B protein?

Introduction section has been expanded according to reviewer’s suggestion. We trust that now is clearer.

  • Can S100B upregulation in maternal blood be due to GDM? Does S100B have any relationship with blood glucose level?

According to reviewer’s suggestion we expanded discussion section adding a paragraph in the study limitations. We trust that now is clearer.

  • Did S100B level reduced following delivery?

Unfortunately, the aims of the present study were to perform a longitudinal S100B monitoring in pregnancy and not in the postnatal period.

  • What are the numbers of each gestational age groups among recruited population? Or all the recruited population of 24-28 GA? Please describe into method section (2.1) and figure 1.

Materials and methods section has been expanded according to reviewer’s suggestion. We trust that now is clearer.

  • Are the collected blood samples treated with anticoagulant? Did you measure S100B from serum or plasma? Please describe clearly on method section (2.2).

Materials and methods S100B measurement section has been expanded according to reviewer’s suggestion. However, Literature data provides evidence that the protein can be collected both in gel-coagulated or EDTA tubes in absence of any side-effects. Anyway we are available to include this information needed.

  • Have you measured amniotic fluid and/or neonatal S100B level? And is there any correlation with maternal blood S100B level? If possible, please include the data.

Unfortunately, the aims of the present study were to perform a longitudinal S100B maternal blood monitoring in pregnancy and not after delivery. Moreover, amniotic fluid assessment in absence of any clinical indications was not performed due to ethical issues.

  • Were the pregnant women with GDM taking insulin or any other medication? Any relationship with maternal blood S100B level? If possible, please include the data.

Materials and Methods and Discussion sections as well have been expanded according to reviewer’s suggestion. We trust that now is clearer.

Reviewer 3 Report

Reviewer Comments for “S100B Maternal Blood Levels in Gestational Diabetes Mellitus 2 are Growth, Gender and Delivery Mode Dependent

In the present study, Abella et al. investigated whether S100B protein is a good neurobiomarker (NBM) to predict the fetal growth using maternal blood. I found this study as an interesting and needed attempt, I have some key comments to be clarified prior to publication,

  1. What is the novel finding in this study compared to previous reports (Ref: 21, 22, 28 and 29)?
  2. Table-1 shows that 36/106 GDM perinatal outcome lead to suspected neurological function. The logical analysis would be focused on these 36 cases and respective S100B protein levels.
  3. A good biomarker needs to have remarkable difference to the baseline (eg. CRP). In case of S100B, the difference is very close between GDM and control group (D4 vs. C4). Moreover, the levels are varying in gestational age. Please provide a line graph with continuous monitoring of each patient (4 data points and 636 lines). Make two color lines, one for GDM and one for controls. This might provide a good visual difference, if any.
  4. Please make significance test for Table-1 (each parameter).
  5. No difference in birth weight between two groups. Is it known the prior-art that GDM perinatal have more weight than normal perinatal? Fig.3 shows a correlation, it would be great to do the analysis for group separately. Similar analysis should be done for the parameters in Table.1 where significance difference noted.
  6. Authors conclude that the present study provided a useful information of S100B protein as a biomarker to predict CNS damage in GDM fetus. However, I don’t see such evidence in the work.

Author Response

We want to thank reviewer for his kind comments and useful suggestions that allowed us to improve the quality of the MS in the present revised form.

  • What is the novel finding in this study compared to previous reports (Ref: 21, 22, 28 and 29)?

Bearing in mind that previous observations by our research group were conducted in different high-risk pregnancies populations, the present series constitute the first observation on S100B measurement in maternal blood of women complicated by GDM. We trust that now is clearer.

  • Table-1 shows that 36/106 GDM perinatal outcome lead to suspected neurological function. The logical analysis would be focused on these 36 cases and respective S100B protein levels.

We want to thank reviewer for his useful suggestion that has been taken into the due account in the present MS revised version. We trust that now is clearer.

  • A good biomarker needs to have remarkable difference to the baseline (eg. CRP). In case of S100B, the difference is very close between GDM and control group (D4 vs. C4). Moreover, the levels are varying in gestational age. Please provide a line graph with continuous monitoring of each patient (4 data points and 636 lines). Make two color lines, one for GDM and one for controls. This might provide a good visual difference, if any.

We re-checked statistical analysis report and we confirm previous results at different monitoring time-points. In order to avoid redundant data figure 2 was uploaded in its original version. However, we are available to change it if needed.

  • Please make significance test for Table-1 (each parameter).

We want to thank reviewer for his suggestion. Table 1 has been expanded as well as results and discussion sections. We trust that now is clearer.

  • No difference in birth weight between two groups. Is it known the prior-art that GDM perinatal have more weight than normal perinatal? Fig.3 shows a correlation, it would be great to do the analysis for group separately. Similar analysis should be done for the parameters in Table.1 where significance difference noted.

We want to thank reviewer for his comment. Table 1 has been revised and typing errors avoided from the text.

  • Authors conclude that the present study provided a useful information of S100B protein as a biomarker to predict CNS damage in GDM fetus. However, I don’t see such evidence in the work.

According to the aforementioned reviewer’s suggestions Conclusions have been expanded. We trust that now is clearer.

Round 2

Reviewer 2 Report

Dear authors, 

Thanks for the revised version of the manuscript. Please see the comments:

  1. Last 2 sentences of abstract are still hard to understand. Please rephrase the sentences. Or you may modify as following: “The present findings show the usefulness of S100B in CNS to monitor high-risk pregnancies during perinatal standard-of-care procedures. Results open the way to study further to investigate its potential role as an early marker of CNS growth/damage in GDM population.
  2. On line 53-54, “…..an acidic calcium binding protein CNS-specific…….” need to write as “……..a CNS-specific acidic calcium binding protein----"
  3. Please rephrase the sentence “In 67 out of 102 106 GDM pregnancies insulin therapy was performed; 34 out of 106 required diet therapy”